# COEVOLVE: A Joint Point Process Model for Information Diffusion and Network Co-evolution

**Mehrdad Farajtabar**[*]    **Yichen Wang**[*]    **Manuel Gomez-Rodriguez**[†]
**Shuang Li**[*]    **Hongyuan Zha**[*]    **Le Song**[*]
Georgia Institute of Technology[*]    MPI for Software Systems[†]
{mehrdad,yichen.wang,sli370}@gatech.edu    manuelgr@mpi-sws.org
{zha,lsong}@cc.gatech.edu

## Abstract

Information diffusion in online social networks is affected by the underlying network topology, but it also has the power to change it. Online users are constantly creating new links when exposed to new information sources, and in turn these links are alternating the way information spreads. However, these two highly intertwined stochastic processes, information diffusion and network evolution, have been predominantly studied *separately*, ignoring their co-evolutionary dynamics.

We propose a temporal point process model, COEVOLVE, for such joint dynamics, allowing the intensity of one process to be modulated by that of the other. This model allows us to efficiently simulate interleaved diffusion and network events, and generate traces obeying common diffusion and network patterns observed in real-world networks. Furthermore, we also develop a convex optimization framework to learn the parameters of the model from historical diffusion and network evolution traces. We experimented with both synthetic data and data gathered from Twitter, and show that our model provides a good fit to the data as well as more accurate predictions than alternatives.

## 1  Introduction

Online social networks, such as Twitter or Weibo, have become large information networks where people share, discuss and search for information of personal interest as well as breaking news [1]. In this context, users often forward to their *followers* information they are exposed to via their *followees*, triggering the emergence of information *cascades* that travel through the network [2], and constantly create new links to information sources, triggering changes in the network itself over time. Importantly, recent empirical studies with Twitter data have shown that both information diffusion and network evolution are coupled and network changes are often triggered by information diffusion [3, 4, 5].

While there have been many recent works on modeling information diffusion [2, 6, 7, 8] and network evolution [9, 10, 11], most of them treat these two stochastic processes independently and separately, ignoring the influence one may have on the other over time. Thus, to better understand information diffusion and network evolution, there is an urgent need for joint probabilistic models of the two processes, which are largely inexistent to date.

In this paper, we propose a probabilistic generative model, COEVOLVE, for the joint dynamics of information diffusion and network evolution. Our model is based on the framework of temporal point processes, which explicitly characterize the continuous time interval between events, and it consists of two interwoven and interdependent components (refer to Appendix B for an illustration):

I. **Information diffusion process.** We design an "identity revealing" multivariate Hawkes process [12] to capture the mutual excitation behavior of retweeting events, where the intensity of such events in a user is boosted by previous events from her time-varying set of followees. Al-

though Hawkes processes have been used for information diffusion before [13, 14, 15, 16, 17, 18, 19], the key innovation of our approach is to explicitly model the excitation due to a particular source node, hence revealing the identity of the source. Such design reflects the reality that information sources are explicitly acknowledged, and it also allows a particular information source to acquire new links in a rate according to her "informativeness".

II. **Network evolution process.** We model link creation as an "information driven" survival process, and couple the intensity of this process with retweeting events. Although survival processes have been used for link creation before [20, 21], the key innovation in our model is to incorporate retweeting events as the driving force for such processes. Since our model has captured the source identity of each retweeting event, new links will be targeted toward the information sources, with an intensity proportional to their degree of excitation and each source's influence.

Our model is designed in such a way that it allows the two processes, information diffusion and network evolution, unfold simultaneously in the same time scale and excise bidirectional influence on each other, allowing sophisticated coevolutionary dynamics to be generated (*e.g.*, see Figure 5). Importantly, the flexibility of our model does not prevent us from efficiently simulating diffusion and link events from the model and learning its parameters from real world data:

- **Efficient simulation.** We design a scalable sampling procedure that exploits the sparsity of the generated networks. Its complexity is $O(nd \log m)$, where $n$ is the number of samples, $m$ is the number of nodes and $d$ is the maximum number of followees per user.
- **Convex parameters learning.** We show that the model parameters that maximize the joint likelihood of observed diffusion and link creation events can be found via convex optimization.

Finally, we experimentally verify that our model can produce coevolutionary dynamics of information diffusion and network evolution, and generate retweet and link events that obey common information diffusion patterns (*e.g.*, cascade structure, size and depth), static network patterns (*e.g.*, node degree) and temporal network patterns (*e.g.*, shrinking diameter) described in related literature [22, 10, 23]. Furthermore, we show that, by modeling the coevolutionary dynamics, our model provide significantly more accurate link and diffusion event predictions than alternatives in large scale Twitter dataset [3].

## 2 Backgrounds on Temporal Point Processes

A temporal point process is a random process whose realization consists of a list of discrete events localized in time, $\{t_i\}$ with $t_i \in \mathbb{R}^+$ and $i \in \mathbb{Z}^+$. Many different types of data produced in online social networks can be represented as temporal point processes, such as the times of retweets and link creations. A temporal point process can be equivalently represented as a counting process, $N(t)$, which records the number of events before time $t$. Let the history $\mathcal{H}(t)$ be the list of times of events $\{t_1, t_2, \ldots, t_n\}$ up to but not including time $t$. Then, the number of observed events in a small time window $dt$ between $[t, t+dt)$ is $dN(t) = \sum_{t_i \in \mathcal{H}(t)} \delta(t-t_i) \, dt$, and hence $N(t) = \int_0^t dN(s)$, where $\delta(t)$ is a Dirac delta function. More generally, given a function $f(t)$, we can define the convolution with respect to $dN(t)$ as

$$f(t) \star dN(t) := \int_0^t f(t - \tau) \, dN(\tau) = \sum_{t_i \in \mathcal{H}(t)} f(t - t_i). \tag{1}$$

The point process representation of temporal data is fundamentally different from the discrete time representation typically used in social network analysis. It directly models the time interval between events as random variables, and avoid the need to pick a time window to aggregate events. It allows temporal events to be modeled in a more fine grained fashion, and has a remarkably rich theoretical support [24].

An important way to characterize temporal point processes is via the conditional intensity function — a stochastic model for the time of the next event given all the times of previous events. Formally, the conditional intensity function $\lambda^*(t)$ (intensity, for short) is the conditional probability of observing an event in a small window $[t, t + dt)$ given the history $\mathcal{H}(t)$, *i.e.*,

$$\lambda^*(t)dt := \mathbb{P}\left\{\text{event in } [t, t + dt) | \mathcal{H}(t)\right\} = \mathbb{E}[dN(t)|\mathcal{H}(t)], \tag{2}$$

where one typically assumes that only one event can happen in a small window of size $dt$, *i.e.*, $dN(t) \in \{0, 1\}$. Then, given a time $t' \geqslant t$, we can also characterize the conditional probability that no event happens during $[t, t')$ and the conditional density that an event occurs at time $t'$

as $S^*(t') = \exp(-\int_t^{t'} \lambda^*(\tau)\,d\tau)$ and $f^*(t') = \lambda^*(t')\,S^*(t')$ respectively [24]. Furthermore, we can express the log-likelihood of a list of events $\{t_1, t_2, \ldots, t_n\}$ in an observation window $[0, T)$ as

$$\mathfrak{L} = \sum_{i=1}^{n} \log \lambda^*(t_i) - \int_0^T \lambda^*(\tau)\,d\tau, \quad T \geqslant t_n. \tag{3}$$

This simple log-likelihood will later enable us to learn the parameters of our model from observed data.

Finally, the functional form of the intensity $\lambda^*(t)$ is often designed to capture the phenomena of interests. Some useful functional forms we will use later are [24]:

**(i) Poisson process.** The intensity is assumed to be independent of the history $\mathcal{H}(t)$, but it can be a time-varying function, *i.e.*, $\lambda^*(t) = g(t) \geqslant 0$;

**(ii) Hawkes Process.** The intensity models a mutual excitation between events, *i.e.*,

$$\lambda^*(t) = \mu + \alpha\kappa_\omega(t) \star dN(t) = \mu + \alpha \sum\nolimits_{t_i \in \mathcal{H}(t)} \kappa_\omega(t - t_i), \tag{4}$$

where $\kappa_\omega(t) := \exp(-\omega t)\mathbb{I}[t \geqslant 0]$ is an exponential triggering kernel, $\mu \geqslant 0$ is a baseline intensity independent of the history. Here, the occurrence of each historical event increases the intensity by a certain amount determined by the kernel and the weight $\alpha \geqslant 0$, making the intensity history dependent and a stochastic process by itself. We will focus on the exponential kernel in this paper. However, other functional forms for the triggering kernel, such as log-logistic function, are possible, and our model does not depend on this particular choice; and,

**(iii) Survival process.** There is only one event for an instantiation of the process, *i.e.*,

$$\lambda^*(t) = g^*(t)(1 - N(t)), \tag{5}$$

where $\lambda^*(t)$ becomes 0 if an event already happened before $t$.

## 3 Generative Model of Information Diffusion and Network Co-evolution

In this section, we use the above background on temporal point processes to formulate our probabilistic generative model for the joint dynamics of information diffusion and network evolution.

### 3.1 Event Representation

We model the generation of two types of events: tweet/retweet events, $e^r$, and link creation events, $e^l$. Instead of just the time $t$, we record each event as a triplet

$$e^r \;\; \text{or} \;\; e^l \;\; := \;\; ( \underset{\text{destination}}{\underset{\uparrow}{u}}, \;\; \overset{\text{source}}{\overset{\downarrow}{s}}, \;\; \underset{\text{time}}{\underset{\uparrow}{t}} ). \tag{6}$$

**For retweet event**, the triplet means that the destination node $u$ retweets at time $t$ a tweet originally posted by source node $s$. Recording the source node $s$ reflects the real world scenario that information sources are explicitly acknowledged. Note that the occurrence of event $e^r$ does *not* mean that $u$ is directly retweeting from or is connected to $s$. This event can happen when $u$ is retweeting a message by another node $u'$ where the original information source $s$ is acknowledged. Node $u$ will pass on the same source acknowledgement to its followers (*e.g.*, "I agree @a @b @c @s"). Original tweets posted by node $u$ are allowed in this notation. In this case, the event will simply be $e^r = (u, u, t)$. Given a list of retweet events up to but not including time $t$, the history $\mathcal{H}_{us}^r(t)$ of retweets by $u$ due to source $s$ is $\mathcal{H}_{us}^r(t) = \{e_i^r = (u_i, s_i, t_i) | u_i = u \text{ and } s_i = s\}$. The entire history of retweet events is denoted as $\mathcal{H}^r(t) := \cup_{u,s \in [m]} \mathcal{H}_{us}^r(t)$.

**For link creation event**, the triplet means that destination node $u$ creates at time $t$ a link to source node $s$, *i.e.*, from time $t$ on, node $u$ starts following node $s$. To ease the exposition, we restrict ourselves to the case where links cannot be deleted and thus each (directed) link is created only once. However, our model can be easily augmented to consider multiple link creations and deletions per node pair, as discussed in Section 8. We denote the link creation history as $\mathcal{H}^l(t)$.

### 3.2 Joint Model with Two Interwoven Components

Given $m$ users, we use two sets of counting processes to record the generated events, one for information diffusion and the other for network evolution. More specifically,

**I**. Retweet events are recorded using a matrix $\boldsymbol{N}(t)$ of size $m \times m$ for each fixed time point $t$. The $(u, s)$-th entry in the matrix, $N_{us}(t) \in \{0\} \cup \mathbb{Z}^+$, counts the number of retweets of $u$ due to source $s$ up to time $t$. These counting processes are "identity revealing", since they keep track of the source node that triggers each retweet. This matrix $\boldsymbol{N}(t)$ can be dense, since $N_{us}(t)$ can be nonzero even when node $u$ does not directly follow $s$. We also let $d\boldsymbol{N}(t) := (\ dN_{us}(t)\ )_{u,s \in [m]}$.

**II**. Link events are recorded using an adjacency matrix $\boldsymbol{A}(t)$ of size $m \times m$ for each fixed time point $t$. The $(u, s)$-th entry in the matrix, $A_{us}(t) \in \{0, 1\}$, indicates whether $u$ is directly following $s$. That is $A_{us}(t) = 1$ means the directed link has been created before $t$. For simplicity of exposition, we do not allow self-links. The matrix $\boldsymbol{A}(t)$ is typically sparse, but the number of nonzero entries can change over time. We also define $d\boldsymbol{A}(t) := (\ dA_{us}(t)\ )_{u,s \in [m]}$.

Then the interwoven information diffusion and network evolution processes can be characterized using their respective intensities $\mathbb{E}[d\boldsymbol{N}(t) \,|\, \mathcal{H}^r(t) \cup \mathcal{H}^l(t)] = \boldsymbol{\Gamma}^*(t)\,dt$ and $\mathbb{E}[d\boldsymbol{A}(t) \,|\, \mathcal{H}^r(t) \cup \mathcal{H}^l(t)] = \boldsymbol{\Lambda}^*(t)\,dt$, where $\boldsymbol{\Gamma}^*(t) = (\ \gamma^*_{us}(t)\ )_{u,s \in [m]}$ and $\boldsymbol{\Lambda}^*(t) = (\ \lambda^*_{us}(t)\ )_{u,s \in [m]}$. The sign $*$ means that the intensity matrices will depend on the joint history, $\mathcal{H}^r(t) \cup \mathcal{H}^l(t)$, and hence their evolution will be coupled. By this coupling, we make: (i) the counting processes for link creation to be "information driven" and (ii) the evolution of the linking structure to change the information diffusion process. Refer to Appendix B for an illustration of our joint model. In the next two sections, we will specify the details of these two intensity matrices.

### 3.3 Information Diffusion Process

We model the intensity, $\boldsymbol{\Gamma}^*(t)$, for retweeting events using multivariate Hawkes process [12]:

$$\gamma^*_{us}(t) = \mathbb{I}[u = s]\,\eta_u + \mathbb{I}[u \neq s]\,\beta_s \sum_{v \in \mathcal{F}_u(t)} \kappa_{\omega_1}(t) \star (A_{uv}(t)\,dN_{vs}(t)), \tag{7}$$

where $\mathbb{I}[\cdot]$ is the indicator function and $\mathcal{F}_u(t) := \{v \in [m] : A_{uv}(t) = 1\}$ is the current set of followees of $u$. The term $\eta_u \geqslant 0$ is the intensity of original tweets by a user $u$ on his own initiative, becoming the source of a cascade and the term $\beta_s \sum_{v \in \mathcal{F}_u(t)} \kappa_\omega(t) \star (A_{uv}(t)\,dN_{vs}(t))$ models the propagation of peer influence over the network, where the triggering kernel $\kappa_{\omega_1}(t)$ models the decay of peer influence over time.

Note that the retweet intensity matrix $\boldsymbol{\Gamma}^*(t)$ is by itself a stochastic process that depends on the time-varying network topology, the non-zero entries in $\boldsymbol{A}(t)$, whose growth is controlled by the network evolution process in Section 3.4. Hence the model design captures the influence of the network topology and each source's influence, $\beta_s$, on the information diffusion process. More specifically, to compute $\gamma^*_{us}(t)$, one first finds the current set $\mathcal{F}_u(t)$ of followees of $u$, and then aggregates the retweets of these followees that are due to source $s$. Note that these followees may or may not *directly* follow source $s$. Then, the more frequently node $u$ is exposed to retweets of tweets originated from source $s$ via her followees, the more likely she will also retweet a tweet originated from source $s$. Once node $u$ retweets due to source $s$, the corresponding $N_{us}(t)$ will be incremented, and this in turn will increase the likelihood of triggering retweets due to source $s$ among the followers of $u$. Thus, the source does *not* simply broadcast the message to nodes directly following her but her influence propagates through the network even to those nodes that do not directly follow her. Finally, this information diffusion model allows a node to repeatedly generate events in a cascade, and is very different from the independent cascade or linear threshold models [25] which allow at most one event per node per cascade.

### 3.4 Network Evolution Process

We model the intensity, $\boldsymbol{\Lambda}^*(t)$, for link creation using a combination of survival and Hawkes process:

$$\lambda^*_{us}(t) = (1 - A_{us}(t))(\mu_u + \alpha_u\,\kappa_{\omega_2}(t) \star dN_{us}(t)) \tag{8}$$

where the term $1 - A_{us}(t)$ effectively ensures a link is created only once, and after that, the corresponding intensity is set to zero. The term $\mu_u \geqslant 0$ denotes a baseline intensity, which models when a node $u$ decides to follow a source $s$ spontaneously at her own initiative. The term $\alpha_u \kappa_{\omega_2}(t) \star dN_{us}(t)$ corresponds to the retweets of node $u$ due to tweets originally published by source $s$, where the triggering kernel $\kappa_{\omega_2}(t)$ models the decay of interests over time. Here, the higher the corresponding retweet intensity, the more likely $u$ will find information by source $s$ useful and will create a *direct* link to $s$.

The link creation intensity $\mathbf{\Lambda}^*(t)$ is also a stochastic process by itself, which depends on the retweet events, and is driven by the retweet count increments $dN_{us}(t)$. It captures the influence of retweets on the link creation, and closes the loop of mutual influence between information diffusion and network topology.

Note that creating a link is more than just adding a path or allowing information sources to take shortcuts during diffusion. The network evolution makes fundamental changes to the diffusion dynamics and stationary distribution of the diffusion process in Section 3.3. As shown in [14], given a fixed network structure $\mathbf{A}$, the expected retweet intensity $\boldsymbol{\mu}_s(t)$ at time $t$ due to source $s$ will depend of the network structure in a highly nonlinear fashion, *i.e.*, $\boldsymbol{\mu}_s(t) := \mathbb{E}[\mathbf{\Gamma}^*_{.s}(t)] = (e^{(\mathbf{A}-\omega_1\mathbf{I})t} + \omega_1(\mathbf{A} - \omega_1\mathbf{I})^{-1}(e^{(\mathbf{A}-\omega_1\mathbf{I})t} - \mathbf{I}))\,\boldsymbol{\eta}_s$, where $\boldsymbol{\eta}_s \in \mathbb{R}^m$ has a single nonzero entry with value $\eta_s$ and $e^{(\mathbf{A}-\omega_1\mathbf{I})t}$ is the matrix exponential. When $t \to \infty$, the stationary intensity $\bar{\boldsymbol{\mu}}_s = (\mathbf{I} - \mathbf{A}/\omega)^{-1}\,\boldsymbol{\eta}_s$ is also nonlinearly related to the network structure. Thus given two network structures $\mathbf{A}(t)$ and $\mathbf{A}(t')$ at two points in time, which are different by a few edges, the effect of these edges on the information diffusion is not just simply an additive relation. Depending on how these newly created edges modify the eigen-structure of the sparse matrix $\mathbf{A}(t)$, their effect can be drastic to the information diffusion.

**Remark 1.** In our model, each user is exposed to information through a time-varying set of neighbors. By doing so, we couple information diffusion with the network evolution, increasing the practical application of our model to real-network datasets. The particular definition of exposure (*e.g.*, a retweet's neighbor) will depend on the type of historical information that is available. Remarkably, the flexibility of our model allows for different types of diffusion events, which we can broadly classify into two categories. In a first category, events corresponds to the times when an information cascade hits a person, for example, through a retweet from one of her neighbors, but she does not explicitly like or forward the associated post. In a second category, the person decides to explicitly like or forward the associated post and events corresponds to the times when she does so. Intuitively, events in the latter category are more prone to trigger new connections but are also less frequent. Therefore, it is mostly suitable to large event dataset for examples those ones generated synthetically. In contrast, the events in the former category are less likely to inspire new links but found in abundance. Therefore, it is very suitable for real-world sparse data. Consequently, in synthetic experiments we used the latter and in the real one we used the former. It's noteworthy that Eq. (8) is written based on the latter category, but, Fig. 7 in appendix is drawn based on the former.

## 4 Efficient Simulation of Coevolutionary Dynamics

We can simulate samples (link creations, tweets and retweets) from our model by adapting Ogata's thinning algorithm [26], originally designed for multidimensional Hawkes processes. However, a naive implementation of Ogata's algorithm would scale poorly, *i.e.*, for each sample, we would need to re-evaluate $\mathbf{\Gamma}^*(t)$ and $\mathbf{\Lambda}^*(t)$, thus, to draw $n$ samples, we would need to perform $O(m^2 n^2)$ operations, where $m$ is the number of nodes.

We designed a sampling procedure that is especially well-fitted for the structure of our model. The algorithm is based on the following key idea: if we consider each intensity function in $\mathbf{\Gamma}^*(t)$ and $\mathbf{\Lambda}^*(t)$ as a separate Hawkes process and draw a sample from each, it is easy to show that the minimum among all these samples is a valid sample from the model [12]. However, by drawing samples from all intensities, the computational complexity would not improve. However, when the network is sparse, whenever we sample a new node (or link) event from the model, only a small number of intensity functions, in the local neighborhood of the node (or the link), will change. As a consequence, we can reuse most of the samples from the intensity functions for the next new sample and find which intensity functions we need to change in $O(\log m)$ operations, using a heap. Finally, we exploit the properties of the exponential function to update individual intensities for each new sample in $O(1)$: let $t_i$ and $t_{i+1}$ be two consecutive events, then, we can compute $\lambda^*(t_{i+1})$ as $(\lambda^*(t_i) - \mu)\exp(-\omega(t_{i+1} - t_i)) + \mu$ without the need to compare all previous events.

The complete simulation algorithm is summarized in Algorithm 2 in Appendix C. By using Algorithm 2, we reduce the complexity from $O(n^2 m^2)$ to $O(nd \log m)$, where $d$ is the maximum number of followees per node. That means, our algorithm scales logarithmically with the number of nodes and linearly with the number of edges at any point in time during the simulation. We also note that the events for link creations, tweets and retweets are generated in a temporally intertwined and inter-

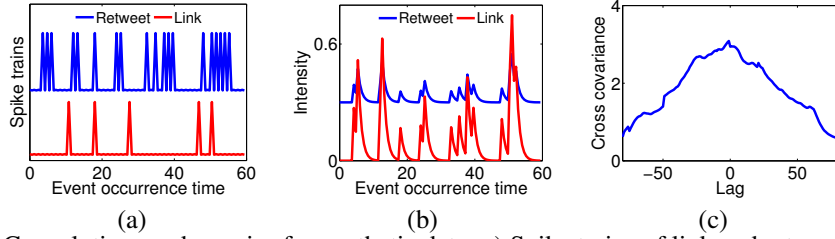

Figure 1: Coevolutionary dynamics for synthetic data. a) Spike trains of link and retweet events. b) Link and retweet intensities. c) Cross covariance of link and retweet intensities.

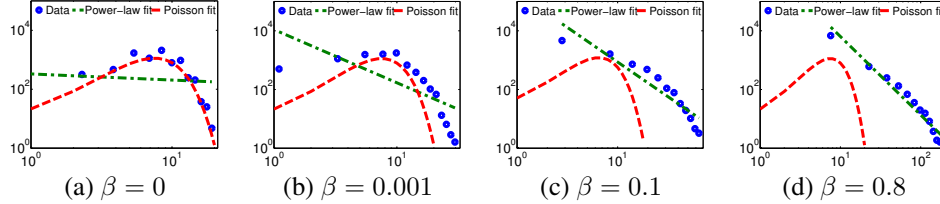

(a) $\beta = 0$     (b) $\beta = 0.001$     (c) $\beta = 0.1$     (d) $\beta = 0.8$

Figure 2: Degree distributions when network sparsity level reaches 0.001 for fixed $\alpha = 0.1$.

leaving fashion by Algorithm 2. This is because every new retweet event will modify the intensity for link creation, and after each link creation we also need to update the retweet intensities.

## 5 Efficient Parameter Estimation from Coevolutionary Events

Given a collection of retweet events $\mathcal{E} = \{e_i^r\}$ and link creation events $\mathcal{A} = \{e_i^l\}$ recorded within a time window $[0, T]$, we can easily estimate the parameters needed in our model using maximum likelihood estimation. Here, we compute the joint log-likelihood $\mathfrak{L}(\{\mu_u\}, \{\alpha_u\}, \{\eta_u\}, \{\beta_s\})$ of these events using Eq. (3), *i.e.*,

$$\underbrace{\sum_{e_i^r \in \mathcal{E}} \log\left(\gamma_{u_i s_i}^*(t_i)\right) - \sum_{u, s \in [m]} \int_0^T \gamma_{us}^*(\tau)\, d\tau}_{\text{tweet / retweet}} + \underbrace{\sum_{e_i^l \in \mathcal{A}} \log\left(\lambda_{u_i s_i}^*(t_i)\right) - \sum_{u, s \in [m]} \int_0^T \lambda_{us}^*(\tau)\, d\tau}_{\text{links}}. \quad (9)$$

For the terms corresponding to retweets, the log term only sums over the actual observed events, but the integral term actually sums over all possible combination of destination and source pairs, even if there is no event between a particular pair of destination and source. For such pairs with no observed events, the corresponding counting processes have essentially survived the observation window $[0, T]$, and the term $-\int_0^T \gamma_{us}^*(\tau)d\tau$ simply corresponds to the log survival probability. Terms corresponding to links have a similar structure to those for retweet.

Since $\gamma_{us}^*(t)$ and $\lambda_{us}^*$ are linear in the parameters $(\eta_u, \beta_s)$ and $(\mu_u, \alpha_u)$ respectively, then $\log(\gamma_{us}^*(t))$ and $\log(\lambda_{us}^*)$ are concave functions in these parameters. Integration of $\gamma_{us}^*(t)$ and $\lambda_{us}^*$ still results in linear functions of the parameters. Thus the overall objective in Eq. (9) is concave, and the global optimum can be found by many algorithms. In our experiments, we adapt the efficient algorithm developed in previous work [18, 19]. Furthermore, the optimization problem decomposes in $m$ independent problems, one per node $u$, and can be readily parallelized.

## 6 Properties of Simulated Co-evolution, Networks and Cascades[*]

In this section, we perform an empirical investigation of the properties of the networks and information cascades generated by our model. In particular, we show that our model can generate co-evolutionary retweet and link dynamics and a wide spectrum of static and temporal network patterns and information cascades. Appendix D contains additional simulation results and visualizations. Appendix E contains an evaluation of our model estimation method in synthetic data.

**Retweet and link coevolution.** Figures 1(a,b) visualize the retweet and link events, aggregated across different sources, and the corresponding intensities for one node and one realization, picked at random. Here, it is already apparent that retweets and link creations are clustered in time and often follow each other. Further, Figure 1(c) shows the cross-covariance of the retweet and link creation intensity, computed across multiple realizations, for the same node, *i.e.*, if $f(t)$ and $g(t)$ are two intensities, the cross-covariance is a function of the time lag $\tau$ defined as $h(\tau) = \int f(t + \tau)g(t)\, dt$. It can be seen that the cross-covariance has its peak around 0, *i.e.*, retweets and link creations are

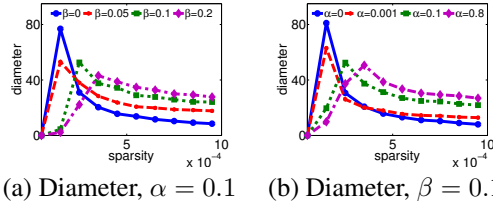

(a) Diameter, $\alpha = 0.1$    (b) Diameter, $\beta = 0.1$

Figure 3: Diameter for network sparsity 0.001. Panels (a) and (b) show the diameter against sparsity over time for fixed $\alpha = 0.1$, and for fixed $\beta = 0.1$ respectively.

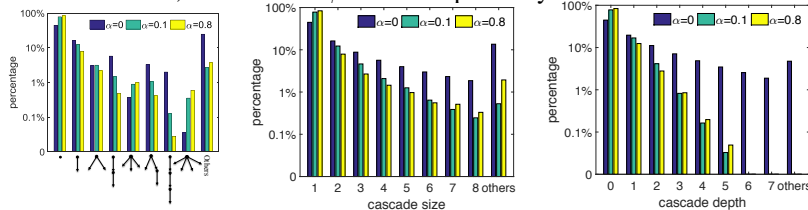

Figure 4: Distribution of cascade structure, size and depth for different $\alpha$ values and fixed $\beta = 0.2$.

highly correlated and co-evolve over time. For ease of exposition, we illustrated co-evolution using one node, however, we found consistent results across nodes.

**Degree distribution.** Empirical studies have shown that the degree distribution of online social networks and microblogging sites follow a power law [9, 1], and argued that it is a consequence of the rich get richer phenomena. The degree distribution of a network is a power law if the expected number of nodes $m_d$ with degree $d$ is given by $m_d \propto d^{-\gamma}$, where $\gamma > 0$. Intuitively, the higher the values of the parameters $\alpha$ and $\beta$, the closer the resulting degree distribution follows a power-law; the lower their values, the closer the distribution to an Erdos-Renyi random graph [27]. Figure 2 confirms this intuition by showing the degree distribution for different values of $\beta$.

**Small (shrinking) diameter.** There is empirical evidence that the diameter of online social networks and microblogging sites exhibit relatively small diameter and shrinks (or flattens) as the network grows [28, 9, 22]. Figures 3(a-b) show the diameter on the largest connected component (LCC) against the sparsity of the network over time for different values of $\alpha$ and $\beta$. Although at the beginning, there is a short increase in the diameter due to the merge of small connected components, the diameter decreases as the network evolves. Here, nodes *arrive* to the network when they follow (or are followed by) a node in the largest connected component.

**Cascade patterns.** Our model can produce the most commonly occurring cascades structures as well as heavy-tailed cascade size and depth distributions, as observed in historical Twitter data [23]. Figure 4 summarizes the results. The higher the $\alpha$ value, the shallower and wider the cascades.

## 7   Experiments on Real Dataset

In this section, we validate our model using a large Twitter dataset containing nearly 550,000 tweet, retweet and link events from more than 280,000 users [3]. We will show that our model can capture the co-evolutionary dynamics and, by doing so, it predicts retweet and link creation events more accurately than several alternatives. Appendix F contains detailed information about the dataset and additional experiments.

**Retweet and link coevolution.** Figures 5(a, b) visualize the retweet and link events, aggregated across different sources, and the corresponding intensities given by our trained model for one node, picked at random. Here, it is already apparent that retweets and link creations are clustered in time and often follow each other, and our fitted model intensities successfully track such behavior. Further, Figure 5(c) compares the cross-covariance between the empirical retweet and link creation intensities and between the retweet and link creation intensities given by our trained model, computed across multiple realizations, for the same node. The similarity between both cross-covariances is striking and both has its peak around 0, *i.e.*, retweets and link creations are highly correlated and co-evolve over time. For ease of exposition, as in Section 6, we illustrated co-evolution using one node, however, we found consistent results across nodes (see Appendix F).

**Link prediction.** We use our model to predict the identity of the source for each test link event, given the historical (link and retweet) events before the time of the prediction, and compare its performance with two state of the art methods, denoted as TRF [3] and WENG [5]. TRF measures

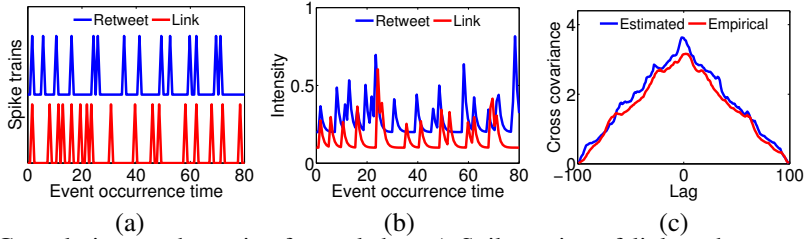

(a)  (b)  (c)

Figure 5: Coevolutionary dynamics for real data a) Spike trains of link and retweet events. b) Estimated link and retweet intensities. c) Empirical and estimated cross covariance of link and retweet intensities

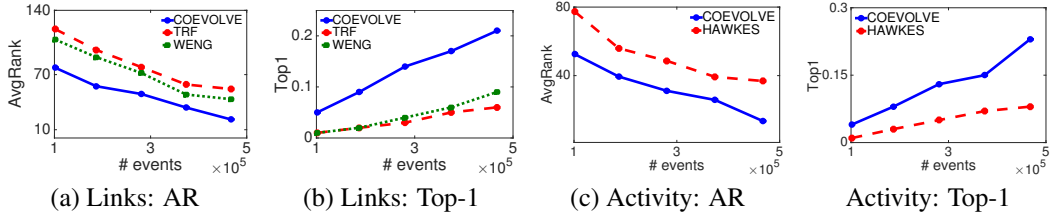

(a) Links: AR    (b) Links: Top-1    (c) Activity: AR    Activity: Top-1

Figure 6: Prediction performance in the Twitter dataset by means of average rank (AR) and success probability that the true (test) events rank among the top-1 events (Top-1).

the probability of creating a link from a source at a given time by simply computing the proportion of new links created from the source with respect to the total number of links created up to the given time. WENG considers different link creation strategies and makes a prediction by combining them.

We evaluate the performance by computing the probability of all potential links using different methods, and then compute (i) the average rank of all true (test) events (AvgRank) and, (ii) the success probability (SP) that the true (test) events rank among the top-1 potential events at each test time (Top-1). We summarize the results in Fig. 6(a-b), where we consider an increasing number of training retweet/tweet events. Our model outperforms TRF and WENG consistently. For example, for $8 \cdot 10^4$ training events, our model achieves a SP 2.5x times larger than TRF and WENG.

**Activity prediction.** We use our model to predict the identity of the node that is going to generate each test diffusion event, given the historical events before the time of the prediction, and compare its performance with a baseline consisting of a Hawkes process without network evolution. For the Hawkes baseline, we take a snapshot of the network right before the prediction time, and use all historical retweeting events to fit the model. Here, we evaluate the performance the via the same two measures as in the link prediction task and summarize the results in Figure 6(c-d) against an increasing number of training events. The results show that, by modeling the co-evolutionary dynamics, our model performs significantly better than the baseline.

# 8  Discussion

We proposed a joint continuous-time model of information diffusion and network evolution, which can capture the coevolutionary dynamics, mimics the most common static and temporal network patterns observed in real-world networks and information diffusion data, and predicts the network evolution and information diffusion more accurately than previous state-of-the-arts. Using point processes to model intertwined events in information networks opens up many interesting future modeling work. Our current model is just a show-case of a rich set of possibilities offered by a point process framework, which have been rarely explored before in large scale social network modeling. For example, we can generalize our model to support link deletion by introducing an intensity matrix $\mathbf{\Xi}^*(t)$ modeling link deletions as survival processes, *i.e.*, $\mathbf{\Xi}^*(t) = (g^*_{us}(t)A_{us}(t))_{u,s\in[m]}$, and then consider the counting process $\mathbf{A}(t)$ associated with the adjacency matrix to evolve as $\mathbb{E}[d\mathbf{A}(t)|\mathcal{H}^r(t) \cup \mathcal{H}^l(t)] = \mathbf{\Lambda}^*(t)\,dt - \mathbf{\Xi}^*(t)\,dt$. We also can consider the number of nodes varying over time. Furthermore, a large and diverse range of point processes can also be used in the framework without changing the efficiency of the simulation and the convexity of the parameter estimation, *e.g.*, condition the intensity on additional external features, such as node attributes.

### Acknowledge

The authors would like to thank Demetris Antoniades and Constantine Dovrolis for providing them with the dataset. The research was supported in part by NSF/NIH BIGDATA 1R01GM108341, ONR N00014-15-1-2340, NSF IIS-1218749, NSF CAREER IIS-1350983.

## Footnotes

* Implementation codes are available at https://github.com/farajtabar/Coevolution

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
