[Supplementary Material]

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

# A  Further related work

The works most closely related to ours are empirical studies of information diffusion and network evolution [33, 34, 5, 3, 4]. Among them, [5] was the first to show experimental evidence the likelihood that a user $u$ starts following a user $s$ increases with the number of messages from $s$ seen by $u$. [3] investigated the temporal and statistical characteristics of retweet-driven connections within the Twitter network and then identified the number of retweets as a key factor to infer such connections. Finally, [4] showed that the Twitter network can be characterized by steady rates of change, interrupted by sudden bursts of new connections, triggered by retweet cascades. They also developed a method to predict which retweets are more likely to trigger these bursts.

However, there are fundamental differences between the above mentioned studies and our work. First, they only characterize the effect that information diffusion has on the network dynamics, but not the bidirectional influence. In contrast, our probabilistic generative model takes into account the bidirectional influence between information diffusion and network dynamics. Second, previous studies are mostly empirical and only make binary predictions on link creation events. For example, [5, 3] predict whether a new link will be created based on the number of retweets; [4] predict whether a burst of new links will occur based on the number of retweets and users' similarity. In contrast, our model can learn parameters from real world data, and predict the precise timing of both diffusion and new link events.

# B    Illustration of the joint dynamics of information diffusion and network evolution

Figure 7 demonstrates the coevolution schematically in a toy social network. Consider node $A$ as a source of information and node $G$ as a potential follower. Node $G$ is first exposed to a piece of information originated from $A$ via a retweet from $B$ at time $t_1$ (Figure 7(a)). The intensity $\lambda_{GA}^*(t)$ of $G$ establishing a link to $A$ increases. Then at time $t_2$, when $B$ retweets another tweet posted by $A$, node $G$ again is exposed to information from $A$ and the intensity $\lambda_{GA}^*(t)$ increases again (Figure 7(b)). After $G$ observes sufficient information from $A$, she finds her a valuable source and thus $G$ decides to follow $A$ directly. This is how information diffusion affects network formation.

On the other side, network evolution will affect information diffusion. After $G$ decides to follow $A$ directly, she opens a new path of information for her downstream nodes. Now, when $G$ retweets tweets originally posted by $A$, node $I$ will be exposed and may retweet them too. Here, the exposure at time $t_3$ and $t_4$ is sufficient for $I$ to retweet a tweet originated by $A$ (Figures 7(c) and 7(d)).

Figure 7: Joint dynamics of information diffusion and network evolution. Blue links form the diffusion paths. Green and blue circles are nodes which see the information. Green circles are those who re-share and propagate the information. Orange circles are nodes unaware of A's post.

# C  Model Simulation

---

**Algorithm 1** Efficient Intensity Computation

---
**Global Variabels:**
Last time of intensity computation: $t$
Last value of intensity computation: $I$
**Initialization:**
$t = 0, I = \mu$
**function** $get\_intensity(t')$
    $I' = (I - \mu) \exp(-\omega(t' - t)) + \mu$
    $t = t', I = I'$
    **return** $I$
**end function**

---

**Algorithm 2** Efficient Simulation

---
**Initialization:**
Initialize the priority queue $Q$
**for** $\forall u, s \in [m]$ **do**
    Sample next link event $e_{us}^l$ from $A_{us}$ (Algorithm 3)
    $Q.insert(e_{us}^l)$
    Sample next retweet event $e_{us}^r$ from $N_{us}$ (Algorithm 3)
    $Q.insert(e_{us}^r)$
**end for**
**General Subroutine:**
$t \leftarrow 0$
**while** $t < T$ **do**
    $e \leftarrow Q.extract\_min()$
    **if** $e = (u, s, t')$ is a retweet event **then**
        Update the history $\mathcal{H}_{us}^r(t') = \mathcal{H}_{us}^r(t) \cup \{e\}$
        **for** $\forall v \ s.t. \ u \rightsquigarrow v$ **do**
            Update event intensity: $\gamma_{vs}(t') = \gamma_{vs}(t'^-) + \beta$
            Sample retweet event $e_{vs}^r$ from $N_{vs}$ (Algorithm 3)
            $Q.update\_key(e_{vs}^r)$
            **if** NOT $s \rightsquigarrow v$ **then**
                Update link intensity: $\lambda_{vs}(t') = \lambda_{vs}(t'^-) + \alpha$
                Sample link event $e_{vs}^l$ from $A_{vs}$ (Algorithm 3)
                $Q.update\_key(e_{vs}^l)$
            **end if**
        **end for**
    **else**
        Update the history $\mathcal{H}_{us}^l(t') = \mathcal{H}_{us}^l(t) \cup \{e\}$
        $\lambda_{us}(t) \leftarrow 0 \ \forall t > t'$
    **end if**
    $t \leftarrow t'$
**end while**

---

**Algorithm 3** Sampling

---
**Input:** Current time: $t$
**Output:** Next event time: $s$
Set $s \leftarrow t$ and $\hat{\lambda} \leftarrow \lambda(s)$ (Algorithm 1)
**while** $s < T$ **do**
    Generate $g \sim exp(\hat{\lambda})$ set $s \leftarrow s + g$
    Set $\bar{\lambda} \leftarrow \lambda(s)$ (Algorithm 1)
    Rejection test: Generate $d \sim \mathcal{U}(0, 1)$
    **if** $d \times \hat{\lambda} < \bar{\lambda}$ **then**   **return** $s$  **else** $\hat{\lambda} = \bar{\lambda}$
**end while**
**return** $s$

---

# D   Additional Experiments on Co-Evolution, Network Properties and Cascade Patterns

**Simulation settings.** Throughout this section, we simulate the evolution of a 8,000-node network as well as the propagation of information over the network by sampling from our model using Algorithm 2. We set the exogenous intensities of the link event and diffusion event intensities to $\mu_u = \mu = 4 \times 10^{-6}$ and $\eta_u = \eta = 1.5$ respectively, and the triggering kernel parameter to $\omega_1 = \omega_2 = 1$. The parameter $\mu$ determines the independent growth of the network, roughly speaking, the expected number of links each user establishes spontaneously before time $T$ is $\mu T$. Whenever we investigate a static property, we choose the same sparsity level of $0.001$.

**Degree Distribution.** Figure 8 shows the degree distribution against the sparsity of the network over time for different values of $\alpha$. For sufficiently high values of $\alpha$, the degree distribution follows a power-law. The lower the $\alpha$ values, the closer the network is to an Erdos-Renyi random graph [27], and as a consequence the closer is the degree distribution to a Poisson. In fact, it is easy to show that our model outputs Erdos-Renyi graphs for $\alpha = 0$.

(a) $\alpha = 0$      (b) $\alpha = 0.05$      (c) $\alpha = 0.1$      (d) $\alpha = 0.2$

Figure 8: Degree distributions when network sparsity level reaches $0.001$ for different $\alpha$ values and fixed $\beta = 0.1$.

**Cascade Patterns.** Figure 9 shows the distribution of the seven most common cascade structure, the cascade size and the cascade depth for different values of $\beta$. The model create the most commonly occurring cascades structures as well as heavy-tailed cascade size and depth distributions, as observed in real diffusion data [23]. The larger the value of $\beta$, the more shallow and wider cascades are.

(a)      (b)      (c)

Figure 9: Distribution, size and depth of cascade structures for different $\beta$ values and fixed $\alpha = 0.8$.

**Clustering coefficient.** Triadic closure [29, 11, 30] has been often presented as a plausible link creation mechanism. However, different social networks and microblogging sites present different levels of triadic closure [31]. Importantly, our method is able to generate networks with different levels of triadic closure, as shown by Figure 10(a-b), where we plot the clustering coefficient [32], which is proportional to the frequency of triadic closure, for different values of $\alpha$ and $\beta$.

**Network Visualization.** Figure 11 visualizes several snapshots of the largest connected component (LCC) of two 300-node networks for two particular realizations of our model, under two different values of $\beta$. In both cases, we used $\mu = 2 \times 10^{-4}$, $\alpha = 1$, and $\eta = 1.5$. The top two rows correspond to $\beta = 0$ and represent one end of the spectrum, *i.e.*, Erdos-Renyi random network. Here, the network evolves uniformly. The bottom two rows correspond to $\beta = 0.8$ and represent the

(a) CC, $\alpha = 0.1$      (b) CC, $\beta = 0.1$

Figure 10: Clustering coefficient for network sparsity 0.001. Panels (a) and (b) show the clustering coefficient (CC) against $\beta$ and $\alpha$, respectively.

other end, *i.e.*, scale-free networks. Here, the network evolves locally, and clusters emerge naturally as a consequence of the local growth. They are depicted using a combination of forced directed and Fruchterman Reingold layout with Gephi[1]. This figure clearly illustrates that by careful choice of parameters we can generate networks with a very different structure.

Furthermore, the retweet events (from others as source) of two nodes, A and B, are demonstrated at the bottom row. These two nodes arrive almost at the same time and establish links to two other nodes. However, node A's followees are more central, therefore, A is being exposed to more retweets. As an expected consequence node A will perform more retweets than B does.

Figure 11: Evolution of a network with $\beta = 0$ (1st and 2nd rows) and $\beta = 0.8$ (3rd and 4th rows) and spike trains of nodes A and B (5th row). Node $A$ is connected to more central nodes, therefore, will perform more retweets (originated by others) than B does.

# E  Experiments on Parameter Estimation

**Experimental Setup.** Throughout this section, we experiment with our model considering $m=400$ nodes. We set the model parameters for each node in the network by drawing samples from $\mu \sim U(0, 0.0004)$, $\alpha \sim U(0, 0.1)$, $\eta \sim U(0, 1.5)$ and $\beta \sim U(0, 0.1)$. We then sample up to 60,000 link and information diffusion events from our model using Algorithm 2 and average over 8 different simulation runs.

**Model Estimation.** We evaluate the accuracy of our model estimation procedure via two measures: (i) the relative mean absolute error (*i.e.*, $\mathbb{E}[|x - \hat{x}|/x]$, MAE) between the estimated parameters ($x$) and the true parameters ($\hat{x}$), (ii) the Kendall's rank correlation coefficient between each estimated parameter and its true value, and (iii) test log-likelihood. Figure 12 shows that as we feed more events into the estimation procedure, the estimation becomes more accurate.

(a) Relative MAE  (b) Rank correlation  (c) Test log-likelihood

Figure 12: Performance of model estimation for a 400-node synthetic network.

# F   Additional Experiments on Real Data

**Dataset Description.**

We use a dataset that contains both link events as well as tweets/retweets from millions of Twitter users [3]. In particular, the dataset contains data from three set of users in 20 days; nearly 8 million tweet, retweet, and link events by more than 8 million users. The first set of users (8,779 users) are source nodes $s$, for whom all their tweet times were collected. The second set of users (77,200 users) are the followers of the first set of users, for whom all their retweet times (and source identities) were collected. The third set of users (6,546,650 users) are the users that start following at least one user in the first set during the recording period, for whom all the link times were collected.

In our experiments, we focus on all events (and users) during a 10-day period (Sep. 21 2012 - Sep. 30 2012) and used the information before Sep 21 to construct the initial social network (original links between users). We model the co-evolution in the second 10-day period using our framework. More specifically, in the coevolution modeling, we have 5,567 users in the first layer who post 221,201 tweets. In the second layer 101,465 retweets are generated by the whole 77,200 users in that interval. And in the third layer we have 198,518 users who create 219,134 links to 1978 users (out of 5567) in the first layer.

**Experimental setup.**   We split events into a training set (covering 85% of the retweet and link events) and a test set (covering the remaining 15%) according to time, *i.e.*, all events in the training set occur earlier than those in the test set. We then use our model estimation procedure to fit the parameters from an increasing proportion of events from the training data.

**Coevolution Behavior.**   Figure 13 demonstrates the link and retweet behavior of four typical users chosen randomly in the real-world dataset. For each user, the left panel (real activity) contains the links to a user and retweets her posts receive in 50 hours, and the right panel (real intensity) shows the corresponding intensity learned in the model. By visual inspection, it is already apparent a links or a retweets usually follow each other, but there are some link or retweet events triggered by exogenous causes.

Figure 13: Link and retweet behavior of 4 typical users in the real-world dataset

**Retweet and Link Event Co-evolution.**   Figure 14 shows the cross correlation of retweet and link intensity for 4 typical users chosen randomly from the Twitter dataset. If $f(t)$ and $g(t)$ are the intensity of the retweets of a user's post and links created to her then the cross-correlation is a function of the time lag $\tau$ defined as $h(\tau) = \int_t f(t+\tau)g(t)\,dt$. The empirical cross-correlation is drawn by interpolating the real events. The estimated cross-correlation is computed by simulating from the model using the parameters learned from the real dataset. As expected, the cross-correlation has its peak around 0, *i.e.*, links and activity are highly correlated, and the empirical cross-correlation coincide with the cross-correlation derived from the intensities in the learned model.

Figure 14: Empirical and simulated crosscorrelation for 4 typical users

To further verify that our model can capture the coevolution, we compute the average value of the empirical cross covariance function, denoted by $m_{cc}$, per user. Intuitively, one could expect that our model estimation method should assign higher $\alpha$ and/or $\beta$ values to users with high $m_{cc}$. Figure 15 confirms this intuition on 1,000 users, picked at random users. Whenever a user has high $\alpha$ and/or $\beta$ value, she exhibits a high cross covariance between her created links and retweets.

Figure 15: Empirical cross correlation versus the learned parameters

**Link and Activity Prediction.** Figure 16 shows the success probability that the true (test) events rank among the top-10 potential events at each test time for both link and activity prediction in the Twitter dataset.

(a) Link Prediction          (b) Activity prediction

Figure 16: Prediction performance in the Twitter dataset by means of the success probability that the true (test) events rank among the top-10 events.

**Test log-likelihood.** We compute the link test log-likelihood using our trained models on the third set of users. Figure 17 summarizes the results, where we consider an increasing number of training retweet/tweet events from the first and second set.

**Model Checking.** Given all $t_i$ and $t_{i+1}$ subsequent event times generated using a Hawkes process, then, by the time changing theorem [12], the intensity integrals $\int_{t_i}^{t_{i+1}} \lambda(t)\,dt$ should conform to the unit-rate exponential distribution. Figure 18 presents the quantiles of the intensity integrals computed using the intensities with the parameters estimated from the real Twitter data against the quantiles of the unit-rate exponential distribution. It clearly shows that the points approximately lie on the same line, giving empirical evidence that a Hawkes process is the right model to capture the real dynamics.

Figure 17: Test link log-likelihood in the Twitter dataset.

(a) Link process       (b) Retweet process

Figure 18: Quantile plots of the intensity integrals from the real link and retweet event time