[Reviews · NeurIPS 2015]

Submitted by Assigned_Reviewer_1

This paper extends previous work on modeling network interactions with multivariate Hawkes processes by introducing a time-varying network into the model. For example, when one Twitter user publishes a tweet, followers of that user are likely to retweet in response. The dynamic network is intended to capture the creation of new connections, for example, when one Twitter user begins to follow another. The instantaneous network is represented by a binary adjacency matrix, and edges are added to the network according to a survival process with a event-driven rate. If one user frequently retweets another user's messages, then it is likely they will begin to follow that user and thereby add a new connection to the network. This combined modeling of activity and network evolution is captured in a joint probabilistic model, and inference is performed by maximum likelihood estimation.

The problem is well-posed and elegantly solved. The paper could be strengthened by addressing a few concerns. First, it appears that the interaction weights ($\beta_s$'s) are shared by all interactions originating from source $s$. Intuitively, it seems that weights should instead be connection specific since the influence of a source node may vary from destination to destination. That is, it seems that $A$ should be accompanied by a weight matrix $W$. Second, this problem is conceptually related to the problem of inferring time-varying connection strengths given observations of spiking activity, which has received particular attention in the context of modeling neural synaptic plasticity. See Stevenson and Kording (NIPS 2011), Linderman, Stock, and Adams (NIPS 2014), and Song et. al. (J. Neuro. Meth. 2015). These papers model the evolution of a real-valued weight matrix rather than a binary adjacency matrix. Third, though in the context of Twitter it may be reasonable to assume that events are marked with their source and destination, in more general scenarios we may not observe the source. Your model could handle this scenario with the addition of latent variables for the unknown source, as in Simma and Jordan (UAI 2011).

Notes: - The legend in Figure 6 says "Hawks" instead of "Hawkes"

- "Twitter" and "Hawkes" should be capitalized in references - Linderman and Adams reference should be to ICML 2014 rather than arxiv

* UPDATE * I appreciate the authors responses. I do think some of those clarifications/generalizations could be worthwhile additions to the paper.
Summary: This is a well-written paper that tackles an interesting problem -- the joint evolution of social network activity and the underlying network structure -- with a novel probabilistic model. The efficacy of the proposed model is demonstrated on real and synthetic data.

Submitted by Assigned_Reviewer_2

As the authors note, there has been a lot of recent work on self-exciting temporal point processes for modeling information diffusion, but prior work considers the dynamics of events on networks and the dynamics of the network itself as two separate processes. The main contribution of this paper, in contrast, is that it jointly models both of these processes: an individual's activity is prompted by the activity of those around them, and the more activity that a given individual is responsible for, the more likely they are to be followed by others.

The authors do a nice job of explaining and mathematizing this relatively simple idea through the language of temporal point processes, and of deriving a method to efficiently simulate data from the model. They simulate different parameter settings to show that the model can reproduce varying degree distributions (from Poisson to power-law), and realistic diameter, clustering coefficient, and cascade size distributions.

Fitting the model is shown to be a concave optimization problem, and the authors use this to model a reasonably sized Twitter corpus. They investigate the co-evoluation of retweets and link creation events, and look at the problem of predicting (the source of) new links and future activity, finding superior performance to past approaches. Overall the results are convincing, although it would be interesting to see how well the method performs on more conventional and well-studied tasks such as predicting the success of future events.

The paper is clearly written, although grammar improvements could be made throughout.

One downside of the paper is that many details are left to the Appendix.
Summary: This paper presents a joint model of how information diffuses throughout a network and how the network evolves over time in response, using a set of coupled temporal point processes to do so. The authors derive efficient methods for sampling from the model and estimating its parameters from observed data, show that the model captures a variety of well-known temporal and structural features of networks, and show good performance in predicting new links and future activity in real data.

Author Feedback
Author rebuttal: Thank you for your careful reading and detailed comments; these will help improve the final version of our paper if accepted.

Reviewer_2:

We agree with the reviewer that \beta can be made connection specific. In the paper, we explain the case where \beta_s is source specific only for simplicity of exposition. If we consider all \beta_{su}, we will have a larger set of parameters for each pair of users with tweet-retweet relationship, but the estimation procedure will remain convex. In fact, if computation and storage is not an issue, one can take into account even more complex scenarios. For instance, the middle user, say v, who is along the path of diffusion and transmits the tweet originated from s to u can also be involved, i.e., defining $\beta_{svu}$ and so on.

We agree that learning time-varying interaction coefficients is interesting by making $beta_{su}$ a function of time. The simulation
would not be affected and the estimation of time-varying interaction can still be carried out via a convex optimization problem in a fashion like [1].

We agree that incorporating latent variables and simultaneously estimating the hidden diffusion network is also interesting, and can make the model more
broadly applicable. One can augment the model in a fashion similar to [2] and [3].

In this work we tried to formalize the idea of co-evolution and articulate its essence, simplicity, and power for modeling processes of and on the network. There are many other extensions to the model. We discussed some of these extensions in the last section of the main paper.

[1] M. Kolar, L. Song, A. Ahmed, and E. P. Xing. Estimating time-varying networks. AOAS 2010.
[2] K. Zhou, H. Zha, L. Song. Learning Social Infectivity in Sparse Low-rank Networks Using Multi-dimensional Hawkes Processes. AISTAT 2013.
[3] S. Linderman, R. Adams. Discovering Latent Network Structure in Point Process Data. ICML, 2014.

Reviewer_3:

In our paper, we included two examples of prediction tasks in our experiments: link and retweet prediction (Figure 6). We agree with the
reviewer that other prediction tasks can be of interest, for example, predicting the time of the next event, the number of events, etc.
However, due to space limitations, we leave the experiments on additional prediction tasks as future work.